# Screening for *Schistosoma* spp. and *Leishmania* spp. DNA in Serum of Ghanaian Patients with Acquired Immunodeficiency

**DOI:** 10.3390/pathogens11070760

**Published:** 2022-07-02

**Authors:** Franziska Weinreich, Felix Weinreich, Andreas Hahn, Ralf Matthias Hagen, Holger Rohde, Fred Stephen Sarfo, Torsten Feldt, Albert Dompreh, Shadrack Osei Asibey, Richard Boateng, Hagen Frickmann, Kirsten Alexandra Eberhardt

**Affiliations:** 1Department of Microbiology and Hospital Hygiene, Bundeswehr Hospital Hamburg, 20359 Hamburg, Germany; franziskaweinreich@bundeswehr.org (F.W.); felixweinreich@bundeswehr.org (F.W.); frickmann@bnitm.de (H.F.); 2Department of Medical Microbiology, Virology and Hygiene, University Medicine Rostock, 18057 Rostock, Germany; andreas.hahn@uni-rostock.de; 3Department of Microbiology and Hospital Hygiene, Bundeswehr Central Hospital Koblenz, 56070 Koblenz, Germany; ralfmatthiashagen@bundeswehr.org; 4Institute of Medical Microbiology, Virology and Hygiene, University Medical Center Hamburg-Eppendorf (UKE), 20251 Hamburg, Germany; rohde@uke.de; 5Department of Medicine, Kwame Nkrumah University of Science and Technology, Kumasi 00233, Ghana; fsarfo.chs@knust.edu.gh; 6Department of Medicine, Komfo Anokye Teaching Hospital, Kumasi 00233, Ghana; shakosbey19@gmail.com; 7Department of Gastroenterology, Hepatology and Infectious Diseases, University Medical Center Düsseldorf, 40225 Düsseldorf, Germany; Torsten.Feldt@med.uni-duesseldorf.de; 8Department of Clinical Microbiology, Komfo Anokye Teaching Hospital, Kumasi 00233, Ghana; adompreh@gmail.com (A.D.); richardboateng166@gmail.com (R.B.); 9Department of Tropical Medicine, Bernhard Nocht Institute for Tropical Medicine & I. Department of Medicine, University Medical Center Hamburg-Eppendorf, 20359 Hamburg, Germany; 10Division of Hygiene and Infectious Diseases, Institute of Hygiene and Environment, 20539 Hamburg, Germany

**Keywords:** schistosomiasis, leishmaniasis, HIV, Ghana, epidemiology, molecular diagnosis

## Abstract

Both *Schistosoma* spp. (species) and *Leishmania* spp. are prevalent in Ghana in West Africa. However, little is known about their local occurrence in immunocompromised individuals. In the study presented here, the real-time PCR-(polymerase chain reaction-)based screening for repetitive DNA (deoxyribonucleotide acid) sequences from the genomes of *Leishmania (L.)* spp. and *Schistosoma (S.)* spp. was performed in the serum of HIV-(human immunodeficiency virus-)infected Ghanaian patients. In 1083 assessed serum samples from HIV-positive and HIV-negative Ghanian patients, *Leishmania* spp.-specific DNA was not detected, while the diagnostic accuracy-adjusted prevalence estimation suggested a 3.6% prevalence of the *S. mansoni* complex and a 0.5% prevalence of the *S. haematobium* complex. Associations of schistosomiasis with younger age, as well as with the male sex, could be shown but not with an HIV status. Weakly significant signals for the associations of schistosomiasis with an increased viral load, reduced CD4+ (CD = cluster of differentiation) T cell count, and a reduced CD4+/CD8+ ratio could be observed but was inconsistently lost in the case of the stratification on the species complex level. So, it is concluded that factors other than HIV status are more likely to have influenced the occurrence of *Schistosoma* spp. infections in the assessed Ghanaian patients. Potential associations between HIV infection-associated factors, such as the viral load and the immune status of the patients, for which weak signals were observed in this hypothesis-forming retrospective assessment, should be confirmed by prospective, sufficiently powered investigations.

## 1. Introduction

Both *Leishmania (L.)* species (spp.) and *Schistosoma (S.)* spp. are eukaryotic parasitic pathogens associated with human diseases prevalent in Ghana. Cutaneous leishmaniasis, in particular, has been described as prevalent in the Ghanaian Ho Municipality in the Volta Region as well as in the Oti Region and the Taviefe community [1,2,3,4,5,6,7,8]. The *Leishmania enriettii* complex could be isolated from the cutaneous lesions of patients from the Ho District [9], while *L. major* and *L. tropica* have been detected in Ghanaian sand flies [10]. The host competence of local rodents and their role in the transmission cycle of *Leishmania* spp. in Ghana is a matter of ongoing academic debate [11]. Although associations between HIV (human immunodeficiency virus) infections and leishmaniasis are considered as well established [12,13], associations between cutaneous leishmaniasis in Ghana and HIV infections are so far poorly characterized.

Focusing on schistosomiasis, both urogenital schistosomiasis, caused by *Schistosoma haematobium* [14,15,16,17,18,19,20,21,22,23,24,25,26], and intestinal schistosomiasis, caused by *S. mansoni* [27,28,29], are common in Ghana regionally, even with high rates of co-infection with both species [27,29]. Pre-school-aged children and school children are affected [30,31,32,33,34] with urogenital schistosomiasis-associated macrohematuria and esoinophilia [35,36]. Seasonal effects exist; in detail, infection rates with *Schistosoma* spp. have been reported to be increased in the rainy season in Ghana [37], and access to safe water sources has been identified as a critical preventive factor [38,39,40,41,42] next to health education, teaching, and training [43,44]. However, educated awareness cannot compensate for lacking resources required for self-protection [21]. Of note, the increased prevalence of schistosomiasis in Ghana was associated with the construction of freshwater dams [45,46].

Similar to leishmaniasis, little is known so far on the epidemiology of schistosomiasis in Ghanaian HIV patients. In a previous assessment by the Komfo Anokye Teaching Hospital in Kumasi, Ghana, a low prevalence of *Schistosoma mansoni* infections within the minor one-digit percent range was recorded in patients with HIV, although an overall increase in the proportion of the intestinal parasite carriage with enteric parasites other than *Schistosoma* spp. compared to HIV-negative individuals was reported [47]. Another study detected increased co-incidence rates of urogenital schistosomiasis and sexually transmitted infections in Ghana [48], but without particular focus on HIV, calling for respective assessments.

For both visceral leishmaniasis and schistosomiasis, real-time PCR (polymerase chain reaction) from the human serum targeting multicopy DNA sequences of the pathogens has been described as a credible diagnostic procedure. For *Leishmania* spp., in particular, real-time PCR targeting kinetoplast DNA (kDNA, DNA = deoxyribonucleotide acid) has been found to be particularly reliable in a recent comparative test assessment [49]. Multicopy target sequences with proven suitability for the diagnosis of schistosomiasis from the patients’ serum comprise *Sm1-7* for the *Schistosoma mansoni* complex and *Dra1* for the *Schistosoma haematobium* complex, respectively, for which a multiplex real-time PCR assay has recently been successfully validated with samples from travel returnees and migrants [50]. Of note, when applied with samples from a Madagascan high endemicity setting for *S. mansoni*, the *Sm1-7* assay proved considerably higher sensitivity compared to the alternative but less repetitive multicopy target ITS2 (internal transcribed spacer) [51].

In this study, sera from Ghanaian HIV-infected patients and a small control group of sera from Ghanaians without HIV were subjected to a real-time PCR-based screening for *Leishmania* spp. [49] and *Schistosoma* spp. [50,51]. Considering the fact that only cutaneous leishmaniasis is known from Ghana [1,2,3,4,5,6,7,8,9,10,11] and the applied *Leishmania* spp.-specific kDNA-PCR assay [49] has been evaluated for visceral leishmaniasis only, the *Leishmania* spp. screening has to be considered experimental. Thereby, it was speculated that immunosuppression associated with an increased replication of the leishmaniae, which cause cutaneous disease, might be associated with a sufficient amount of pathogen DNA in patient sera to be measurable in peripheral serum as well but without evidence for this assumption. Pathogen findings and the recorded cycle threshold (Ct) values within this screening approach were tested for a correlation with the immunological status of the patients. By doing so, the exploratory, hypothesis-forming study intended to contribute to a better understanding of the epidemiology of leishmaniasis and schistosomiasis in Ghanaian HIV patients. One underlying hypothesis was that HIV infection-associated immunosuppression might lead to an increased immunotolerance against parasitic infections and thus to an increased detection rate of freely circulating parasite DNA in the peripheral blood of the patients. In line with this, a higher *Schistosoma* spp. infection intensity was found to be correlated with HIV infection in a Tanzanian assessment [52]. Also, previous evidence suggests that *Schistosoma* spp. infections, particularly in the case of urogenital schistosomiasis, increase the risk of HIV acquisition, as shown in Tanzania, Uganda, and Zimbabwe [53,54,55], while HIV infections do not seem to be independent risk factors for the acquisition of schistosomiasis [56,57].

## 2. Results

### 2.1. Qualitative and Quantitative PCR Results

From 1083 assessed serum samples, four samples had to be excluded from the further assessment due to PCR inhibition, resulting in 1079 included samples. A total of thirty-six (3.4%) samples tested positive for the *S. mansoni* complex and five (0.5%) for the *S. haematobium* complex. Leishmanial DNA was not detected. Applying the known diagnostic accuracy estimations [50] for the *S. mansoni* complex PCR (sensitivity: 0.959, specificity: 0.973) and the *S. haematobium* complex PCR (sensitivity: 0.933, specificity: 1) for the calculation of the diagnostic accuracy-adjusted prevalence conducting the Rogan&Gladen/Gart&Buck prevalence estimator as described recently [58,59,60], adjusted prevalence values of 3.6% for the *S. mansoni* complex and 0.5% for the *S. haematobium* complex were estimated. A semi-quantitative assessment of the recorded cycle threshold (Ct) values resulted in a mean value of 31.6 (standard deviation SD: 2.2, minimum: 24.1, maximum: 35.7) for the *S. mansoni* complex and a mean value of 31.6 (SD: 1.4, minimum: 30.1, maximum: 34.1) for the *S. haematobium* complex.

### 2.2. Associations of PCR Positivity with Age, Sex, and HIV Status

Younger age was associated with positive PCR results for *Schistosoma* spp. on the genus level as well as for the *S. haematobium* complex but not with positive PCR results for the *S. mansoni* complex. Associations between infections and sex could be shown for *Schistosoma* spp. on the genus level, as well as for the *S. mansoni* complex with a higher infection rate in the male sex, but not for the *S. haematobium* complex. With all *Schistosoma mansoni* complex infections occurring in HIV-positive patients, there was a weak significance for an association between *S. mansoni* complex infections and HIV positivity. However, such links were neither detectable for the *S. haematobium* complex nor *Schistosoma* spp. on the genus level. The details are provided in Table 1.

### 2.3. Associations of PCR Positivity with HIV Viral Load, CD4+ (CD = Cluster of Differentiation) T Cell Count, and CD4+/CD8+ Ratio

PCR positivity for *Schistosoma* spp. on the genus level, as well as for the *S. haematobium* complex but not for the *S. mansoni* complex, was weakly associated with a higher HIV viral load. Associations of reduced CD4+ T cell counts with PCR positivity for *Schistosoma* spp. on the genus level, for the *S. mansoni* complex, and for the *S. haematobium* complex were observed. Finally, a lower CD4+/CD8+ ratio was associated with the positivity of PCR for the *Schistosoma haematobium* complex spp., while the calculated weak significance was lost on the genus level and for the *S. mansoni* complex. The details are provided in Table 2.

### 2.4. Correlations of Cycle Threshold (Ct) Values with HIV Viral Load and CD4+ Cell Count

During the correlation of the cycle threshold (Ct) values and HIV viral load, as well as the CD4+ T cell count, a significant positive correlation could be shown for the HIV viral load and the Ct values of the *S. haematobium* complex only. The details are provided in Table 3.

## 3. Discussion

The study was performed to search for associations between systemic *Schistosoma* spp. infections and *Leishmania* spp. infections and the HIV status in a Ghanaian study population with a high proportion of HIV infections. As expected for Ghana, the majority of individuals were free of active schistosomiasis, as indicated by the serum PCR, and no hints of the DNA of systemically circulating leishmaniae were found. As similarly seen in a previous study [47], a low portion of schistosomiasis within the one-digit percent range was observed, and only a weak association of HIV positivity and active *S. mansoni* complex infections could be demonstrated, which disappeared as soon as the results of the *S. haematobium* complex screening were added. Consequently, the HIV status is—if any—only a weak predictor of *Schistosoma* spp. infections in Ghana, even weaker than the observed association with the male sex and younger age. This is well in line with the previous results from Angola, Kenya, and Zambia [56] and quite plausible, as there is no obvious reason why the mode of infection in the case of schistosomiasis should be influenced by the immunological status of the individuals at risk, although one might speculate that poor socio-economic conditions might predispose them for higher risks of acquiring HIV and schistosomiasis.

Focusing on HIV-specific infection- and immune parameters, tendencies of a higher viral load, lower CD4+ T cell count, and lower CD4+/CD8+ ratio with schistosomiasis were observed. However, the significance is low and not consistent for both *Schistosoma* species complexes. The observed positive correlation between the cycle threshold values of the *S. haematobium* complex PCR and the HIV viral load, suggesting associations between low parasitemia and high viremia, makes the interpretation of those results difficult as well. The study is in line with the previous results suggesting higher CD8+ cell counts in HIV patients co-infected with helminths [61,62], as indicated by the recorded lower CD4+/CD8+ ratio. The observed lower CD4+ T cell count, however, is in contrast to the previous studies indicating either comparable [62,63] or even higher CD4+ cell counts [64] in HIV patients co-infected with schistosomes. Also, the findings of the present study are in contrast with a previous study which did not find any hints for higher HIV virus loads in patients with helminth infections [65], however, without a specific focus on schistosomes alone. Considering the partly weak significance levels, as observed in the here-described exploratory assessment, the relevance of these findings is, however, questionable, and confirmation by sufficiently powered confirmatory assessments is required.

Altogether, the observed Ct values were not in an unexpected range. In a recent screening with a Madagascan population, in which the HIV rate was extremely low [66], quite similar Ct values for the *S. mansoni* complex were observed by applying the same real-time PCR assay [51]. Although other factors, such as the worm load in the patients’ blood, also affect the measures of the Ct values, it seems justified that a relevant effect of HIV positivity on the measured Ct values is not very likely. This finding is in contrast with a previous assessment which suggested higher *Schistosoma* spp. infection intensities in HIV-positive individuals in Tanzania [52]. A possible explanation for the discrepancy might comprise socio-economic differences in Ghanaian and Tanzanian HIV patients, potentially resulting in a stronger poverty-related need for risk exposure to schistosome-contaminated surface water by Tanzanian HIV patients. However, the available data are insufficient to verify or falsify this hypothesis.

The study has a number of limitations. Due to the known uneven geographical distribution of schistosomiasis in Ghana [14,15,16,17,18,19,20,21,22,23,24,25,26,27,28,29], the resulting low overall prevalence made any conclusions on significant associations challenging. However, big differences would most likely not have gone undetected. Second, the medical history of deworming therapy of the study population was not available. The frequency of previous therapeutic interventions might be a difficult-to-control source of bias because PCR becomes negative if active infection is therapeutically cured, leading to a lower prevalence estimation of previous infections. To estimate the relevance of this bias, a parallel serological assessment of the samples, as performed in a previous study [51], would have been an option because serology also indicates previous, already successfully cured infections. This was, however, unfeasible due to the financial restrictions of this investigator-initiated study. Third, the uneven proportion of HIV-positive and HIV-negative study participants was a random effect resulting from the composition of the previous studies from which the assessed residual sample materials were taken for this retrospective assessment [52,53]. Accordingly, no case number estimations for the proof of the assumed effects could be performed, and the study has to be considered hypothesis-forming. Fourth, parasitic infections, such as schistosomiasis, are affected by external influences, such as seasonality, as stated above, which has not been specifically addressed in this study. However, (a) the study did not discriminate between acute and chronic infections, and (b) all of the compared groups were exposed to similar exposition conditions, which limits the relevance of such influences. Fifth, the samples’ age of 10 years at the time of the assessment may have resulted in DNA degradation and—associated with this—in an underestimation of the prevalence of the assessed parasitic infections. However, as optimal storage conditions were ensured by the deep-freezing of the samples at −80 °C, the quantitative dimension of this problem was considered to be low. Sixth, due to the hypothesis-forming, exploratory approach of the study and the restricted number of available datasets, the bio-statistic assessments were restricted to simple statistic approaches. In line with this, a standardized significance threshold of a *p* ≤ 0.05 was accepted for each hypothesis, thus accepting that many significances were weak and would have lost presence if correcting for multiple testing, e.g., by applying the Bonferroni–Holm method, would have been applied. Larger-sized and, thus, sufficiently powered confirmatory studies will be necessary to confirm or exclude the observed hypotheses. The data as described in this study may serve for inclusion in sample size calculations depending on the chosen endpoint of such confirmatory assessments.

## 4. Materials and Methods

### 4.1. Patients and Patient Samples

The study was performed with a group of 1095 HIV-positive Ghanaian patients and a smaller comparison group of 107 HIV-negative Ghanaian blood donors. Residual serum samples that had been taken in the course of the previous studies, as detailed elsewhere [67,68,69,70,71], were available for 1083 individuals (980 HIV positive and 103 HIV negative). Clinical information comprised age, sex, HIV status, and the HIV patients’ immunological situation as expressed by the CD4+ lymphocyte count, the CD4+/CD8+ ratio, and the HIV viral load of the HIV-positive patients from the previous assessments [67,68]. Serum samples had been stored at −80 °C prior to the analyses.

### 4.2. Diagnostic Procedures

Nucleic acid extraction was performed by applying the EZ1 Virus Mini Kit v2.0 Kit (Qiagen, Hilden, Germany) on an automated EZ1 Advanced system (Qiagen, Hilden, Germany), an approach that had already allowed the successful detection of *Leishmania* spp. kDNA in serum samples of patients with visceral leishmaniasis [49]. The extractions were performed directly prior to the PCR assessments after 10 years of deep-frozen storage of the serum samples at −80 °C. Afterward, the eluates were assessed by *Leishmania* spp.-specific kDNA real-time PCR [49] and by a triplex real-time PCR targeting the *S. mansoni* complex-specific *Sm1-7* sequence, the *S. haematobium* complex-specific *Dra1* sequence [50], and a Phocid herpes virus (PhHV) sequence that was added as a plasmid to the samples prior to nucleic acid extraction, thus allowing combined extraction and inhibition control assessment as described before [72]. Both PCR assays were run on RotorGene Q cyclers (Qiagen, Hilden, Germany) or magnetic induction cyclers (MIC, Bio Molecular Systems Ltd., London, UK) exactly as detailed before with open access published protocols [49,50]. Each run was accompanied by a plasmid-based positive control and a PCR-grade water-based negative control, also as described in [49,50]. By applying diagnostic conditions identical to the laboratory’s diagnostic routine, standardized quality was ensured.

### 4.3. Statistical Assessment

First, the recorded prevalence of *Leishmania* spp., the *S. mansoni* complex, and the *S. haematobium* complex was assessed for the population and associated with age, sex, HIV status, and immunological status in a descriptive assessment. Second, a diagnostic accuracy-adjusted prevalence estimation was conducted, as described elsewhere [60,73]. Third, a correlation between the immunologically relevant parameters, i.e., the CD4+ lymphocyte count and HIV virus load, on the one hand and the recorded pathogen-specific cycle threshold (Ct) values on the other hand was attempted. The calculations were conducted using the software Stata/IC 15.1 for Mac 64-bit Intel (College Station, TX, USA).

## 5. Conclusions

In spite of the above-mentioned limitations, the assessment provided the first insights into potential associations between schistosomiasis and HIV status in Ghana. The results suggest only weak associations, making the influence of factors other than the HIV status on schistosomiasis highly likely. Weakly significant signs of potential associations between schistosomiasis and unfavorable factors, such as a high viral load and reduced CD4+ T cell count, have to be considered as hypothesis-forming only and should be re-assessed by sufficiently powered prospective studies. This is particularly the case because a recent review only partially confirmed the results of the assessment [57], while, however, the overall evidence is still low.

## Figures and Tables

**Table 1 pathogens-11-00760-t001:** Associations of *Schistosoma* spp. with age, sex, and HIV (human immunodeficiency virus) status of the cohort. Significance was calculated by applying a student’s *t*-test and Fisher’s two-sided exact test. Above each assessed parameter, the numbers of cases and percentages are given, for which the respective parameter was recorded. The total numbers, *n*, differ from 1079 due to partly incomplete datasets. *p*-values ≤ 0.05 were considered as indicators of statistical significance of differences between the compared groups.

	*S.* *mansoni* Positive	*S.* *mansoni* Negative	*p*-Value	*S.* *haematobium* Positive	*S. haematobium* Negative	*p*-Value	*S. mansoni* and/or *S. haematobium* Positive	*S. mansoni* and/or *S. haematobium* Negative	*p*-Value
*n* (%)	36 (3.4)	1014 (96.6)		5 (0.5)	1045 (99.5)		41 (3.9)	1009 (96.1)	
Age in years ± SD	37.1 ± 7.2	39.6 ± 10.0	0.1432	27.2 ± 4.8	39.6 ± 9.9	0.0040	35.9 ± 7.7	39.7 ± 10.0	0.0165
*n* (%)	36 (3.4)	1015 (96.6)		5 (0.5)	1046 (99.5)		41 (3.9)	1010 (96.1)	
Male, *n* (%)	17 (47.22)	245 (24.14)	0.003	2 (40.00)	260 (24.86)	0.603	19 (46.34)	243 (24.06)	0.003
*n* (%)	36 (3.4)	1036 (96.6)		5 (0.5)	1067 (99.5)		41 (3.8)	1031 (96.2)	
HIV positive, *n* (%)	36 (100.00)	933 (90.06)	0.042	3 (60.00)	966 (90.53)	0.075	39 (95.12)	930 (90.20)	0.420

*n* = number; SD = standard deviation.

**Table 2 pathogens-11-00760-t002:** Associations of *Schistosoma* spp. (species) with the HIV (human immunodeficiency virus) viral load, CD 4+ (CD = cluster of differentiation) T cell count, and CD4+/CD8+ ratio of the HIV-positive patients. Significance was calculated by applying the Wilcoxon rank-sum (Mann–Whitney) testing. Above each assessed parameter, the number of cases and percentages are given, for which the respective parameter was recorded. *p*-values ≤ 0.05 were considered as indicators of statistical significance of the differences between the compared groups.

	*S. mansoni* Positive	*S. mansoni* Negative	*p*-Value	*S. haematobium* Positive	*S. haematobium* Negative	*p*-Value	*S. mansoni* and/or *S. haematobium* Positive	*S. mansoni* and/or *S. haematobium* Negative	*p*-Value
*n* (%)	35 (3.8)	886 (96.2)		3 (0.3)	918 (99.7)		38 (4.1)	883 (95.9)	
CD4+ T cell count/µL, median (IQR)	280.0 (69.0–397.0)	398.0 (196.0–623.0)	0.0017	47.0 (8.0–66.0)	392.5 (189.0–610.0)	0.0094	244.0 (58.0–392.0)	398.0 (197.5–629.0)	0.0002
*n* (%)	27 (4.2)	622 (95.8)					30 (4.6)	619 (95.4)	
CD4+/CD8+ T cell ratio, median (IQR)	0.30 (0.17–0.81)	0.42 (0.21–0.75)	0.3277	0.1 (0.0–0.1)	0.4 (0.2–0.7)	0.013	0.25 (0.14–0.50)	0.42 (0.21–0.75)	0.0837
*n* (%)	34 (3.9)	841 (96.1)		3 (0.3)	872 (99.7)		37 (4.2)	838 (95.8)	
Viral load, log10 copies/mL, median (IQR)	4.8 (2.0–5.5)	3.9 (1.6–5.2)	0.1263	5.5 (5.2–6.1)	4.0 (1.6–5.2)	0.0374	5.0 (3.1–5.5)	3.9 (1.6–5.2)	0.0382

*n* = number. IQR = interquartile range.

**Table 3 pathogens-11-00760-t003:** Correlations (Spearman’s rho) of cycle threshold (Ct) values of the real-time PCRs targeting the *S. mansoni* complex and *S. haematobium* complex with the HIV (human immunodeficiency virus) viral load and CD4+ (CD = cluster of differentiation) cell count.

	CD4+ T Cell Count *n*, rho, *p* Value	Viral Load *n*, rho, *p* Value
*S. mansoni* complex specific real-time PCR	35, 0.1521, 0.3830	34, −0.4697, 0.0051
*S. haematobium* complex specific real-time PCR	5, −0.8, 0.1041	3, 1.0000, <0.001

*n* = total number. rho = Speraman’s rho indicating correlation.

## Data Availability

All relevant details are provided in the manuscript. Raw data can be made available at reasonable request.

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
