# Peer review of "Screening for Schistosoma spp. and Leishmania spp. DNA in Serum of Ghanaian Patients with Acquired Immunodeficiency"

_pathogens, 2022, doi:10.3390/pathogens11070760_

Round 1

Reviewer 1 Report

The manuscript of Weinreich et al “Screening for Schistosoma spp. and Leishmania spp. DNA in serum of Ghanaian patients with acquired immunodeficiency” describes the detection of Schistosoma spp and Leishmania spp infection among Ghanaian HIV-patients. The authors did not detect Leishmania spp DNA but Schistosoma spp in serum of HIV patients. They also investigated any potential correlation between the Schistosoma spp infection and the age, sex and the HIV status of the patients but did not observe any clear evidence. The manuscript is well-written with minor comments:

Tables 1 and 2 can be improved. It is quite challenging to read as the width of the columns are small. Increase the width so the name and number can be easily read.

In Table 1, n does not correspond to 1079 samples as described in line 145 and n varies in all rows. Why it that so?

Line 193 S. haemaobium should be written in italic.

Author Response

Dear Reviewers, Dear Editor,

Thank you very much for your time reviewing our manuscript “Screening for Schistosoma spp. and Leishmania spp. DNA in serum of Ghanaian patients with acquired immunodeficiency”. We highly appreciate your constructive suggestions and comments, which contributed to increase the quality of our manuscript. We are pleased to send you a revised version of our article in which we paid particular attention to every reviewer remark. Please find our responses below.

Reviewer 1

Reviewer’s comment #1:

Tables 1 and 2 can be improved. It is quite challenging to read as the width of the columns are small. Increase the width so the name and number can be easily read.

Response to the reviewer’s comment #1:

We agree with the reviewer and adapted the layout of the tables accordingly.

Reviewer’s comment #2:

In Table 1, n does not correspond to 1079 samples as described in line 145 and n varies in all rows. Why it that so?.

Response to the reviewer’s comment #2:

To avoid this confusion, we have clarified in the table headline that the numbers n differ from 1,079 due to partly incomplete datasets.

Reviewer’s comment #3:

Line 193 S. haemaobium should be written in italic.

Response to the reviewer’s comment #3:

Corrected.

Reviewer 2

Reviewer’s comment #1:

The study seemed predicated on the hypothesis that there may be a relationship between HIV status and infection with these parasites, but it’s not clear what the rationale is for that. There is an emphasis that it has not been examined before. However, if schistosomes commonly infect plenty of people who are not immune-compromised, why would a strong association be expected?

Response to the reviewer’s comment #1:

As requested, we have explained at the end of the introduction that one underlying hypothesis of the study concept was that HIV infection-associated immunosuppression might lead to increased immunotolerance against parasitic infections and thus to an increased detection rate of freely circulating parasite DNA in peripheral blood of the patients.

Reviewer’s comment #2:

In addition, the HIV- group was small. Can such a data set really test for an association?

Response to the reviewer’s comment #2:

To avoid this misunderstanding, we have now clarified in the last-but-one sentence of the introduction that the study was performed as a hypothesis-forming exploratory assessment, as it was not powered for the proof of associations. Please also see our response to your 31st comment regarding this issue.

Reviewer’s comment #3:

Two studies show high relapse rates of visceral leishmaniasis in HIV+ people. No studies of cutaneous. Isn’t relapse rate a different question than association with immunocompromised status? Did these studies find no association with visceral leishmaniasis absence/presence and immune status?

Response to the reviewer’s comment #3

We agree that our mentioning of relapse of visceral leishmaniasis in HIV-infected individuals might be confusing as it has nothing to do with the topical focus of our study. Accordingly, we have removed this hint from the last sentence of the first paragraph of the introduction and just stated the general association between visceral leishmaniasis and immune status.

Reviewer’s comment #4:

Infection rates in Ghana go up during rainy season. Were these samples gathered in the rainy season? Seems like the relevance of this statement needs to be dealt with once it is included.

Response to the reviewer’s comment #4:

As suggested, we have now addressed this topic as a new fourth limitation of the study in the last paragraph of the discussion. There, we have not stated that parasitic infections like schistosomiasis are affected by external influences such as seasonality, which has not been specifically addressed in the study. However, a) the study did not discriminate between acute and chronic infections and b) all compared groups were exposed to similar exposition conditions, which limits the relevance of such influences.

Reviewer’s comment #5:

68-69: Change to, “However, education and increased awareness cannot…”

Response to the reviewer’s comment #5:

Corrected.

Reviewer’s comment #6:

70: Change to, “Of note,…”

Response to the reviewer’s comment #6:

Done.

Reviewer’s comment #7:

72: Change to, “Similar to leishmaniasis, little is known…”

Response to the reviewer’s comment #7:

Changed, as advised.

Reviewer’s comment #8:

One study found an overall increase of intestinal parasite carriage. What does this mean? (other parasites, intensity of schistosomes?, intensity of other parasites?). Another study detected increased co-incidence rates of urogenital schisto and STDs.

Response to the reviewer’s comment #8:

To resolve this misunderstanding, we have made it clear that enteric parasites other than Schistosoma spp. were meant.

Reviewer’s comment #9:

What about other countries? Maybe the rationale for the study would be more clear if there were other studies in other populations that found significant associations? While I appreciate that human populations are very different in genetics and culture, especially in Africa, I’m not sure why only literature with data from Ghana are included.

Response to the reviewer’s comment #9:

The introduction is meant to make the reader familiar with the epidemiological situation in Ghana regarding the assessed parasites. However, previous literature on associations with HIV (e.g. references 12 and 13 dealing with the general association of HIV and at least visceral leishmaniasis) have already been including. For schistosomiasis, however, respective assessments are simply missing so far, which explains both their lacking quotation in the introduction and our interest in dealing with this research question.

Reviewer’s comment #10:

96: Isn’t this the basis of the qPCR test itself? Not sure of the purpose of this statement

Response to the reviewer’s comment #10:

Indeed, the sentence had erroneously been phrased in a way which might provoke this misunderstanding. We have rephrased it, now stating that it was speculated that immunosuppression-associated, increased replication of leishmaniae causing cutaneous disease might be associated with a sufficient amount of pathogen DNA in patient sera to be measurable in peripheral serum as well. The non-unambiguous term “amplification” has been replaced.

Reviewer’s comment #11:

99: tested for correlation? You don’t correlate data, you test for a correlation.

Response to the reviewer’s comment #11:

We agree with the reviewer and changed the sentence accordingly.

Reviewer’s comment #12:

105: why aren’t the numbers of each group given here in the text? The HIV+ group was small

Response to the reviewer’s comment #12:

As requested, it has now been clarified in the Methods section how many of the initially included 1083 sera had been taken from HIV-positive and HIV-negative patients.

Reviewer’s comment #13:

110: how was the HIV viral load measured in the previous studies, by qPCR?

Response to the reviewer’s comment #13:

The HIV viral load was measured using the Real-Time HIV-1 PCR system (Abbott Diagnostics, Wiesbaden, Germany).

Reviewer’s comment #14:

111: how long had the samples been stored prior to analysis? Could this be a factor in the mostly negative results? Were the extractions new for this study?

Response to the reviewer’s comment #14:

As requested, we have clarified that the extractions were performed directly prior to the PCR assessments after 10 years of deep-frozen storage of the serum samples at -80°C. As a new with point of the limitations paragraph of the study, we have now additionally stated that the samples' age of 10 years at the time of the assessment may have resulted in DNA degradation and – associated with this – in an underestimation of the prevalence of the assessed parasitic infections. However, as optimal storage conditions were ensured by deep-freezing of the samples at -80°C, the quantitative dimension of this problems was considered to be low.

Reviewer’s comment #15:

2.4: shouldn’t these ethics statements just be included at the end of the paper?

Response to the reviewer’s comment #15:

We agree with the reviewer and deleted section 2.4 from the main text.

Reviewer’s comment #16:

Overall, I am nervous about the number of statistical comparisons being made and that most have at least one category with a small sample size. If one makes enough statistical comparisons, sooner or later there will be one or some that appear significant. It’s not clear that has been considered or corrected for.

Response to the reviewer’s comment #16:

We agree that is an issue of undeniable relevance. Accordingly, we have explained our choice of a simple statistic approach in a new 6th paragraph of the discussion. There, we have stated that due to the hypothesis-forming, exploratory approach of the study and the restricted number of available datasets, the bio-statistic assessments were restricted to simple statistic approaches. In line with this, a standardized significance threshold of p≤0.05 was accepted for each hypothesis, thus accepting that many significances were weak and would have lost presence if correction for multiple testing, e.g., by applying the Bonferroni-Holm method, would have been applied. Larger sized studies will be necessary to confirm or exclude the observed hypotheses.

Reviewer’s comment #17:

I also wonder if it would be more appropriate to use some kind of statistical model to evaluate this data. I am not a statistician, but it seems like this is now often recommended.

Response to the reviewer’s comment #17:

As already stated in response to your 16th comment, we have addressed this topic by including a new sixth limitation in the limitations paragraph of the discussion, stating there that that due to the hypothesis-forming, exploratory approach of the study and the restricted number of available datasets, the bio-statistic assessments were restricted to simple statistic approaches. From our point of few, a more complex statistical approach would have provoked an inadequate sense of certainty which cannot be provided by our hypothesis-forming approach.

Reviewer’s comment #18:

Was the viral load data inclusive of HIV negative patients, or were only HIV+ patient data used?

Response to the reviewer’s comment #18:

The determination of the HIV viral load was only performed in samples of HIV positive participants. As requested, this has now been clarified (methods chapter, sub-heading “Patients and patient samples”, 2nd sentence).

Reviewer’s comment #19:

Tables 1 and 2: It’s not clear what the numbers are underneath the p-values.

Response to the reviewer’s comment #19:

To answer this question, we have clarified that p-values ≤ 0.05 were considered as indicators of statistical significance of differences between the compared groups. Regarding the associated uncertainty, please also see our response to your 16th comment.

Reviewer’s comment #20:

Line 199: ‘Similar as suggested by’ is awkward. Try ‘Like a previous study’, or ‘As similarly seen in a previous study’.

Response to the reviewer’s comment #20:

The phrasing was changed as suggested by the reviewer.

Reviewer’s comment #21:

Line 202: which got lost is not appropriate terminology

Response to the reviewer’s comment #21:

We agree and used the wording “disappeared” instead.

Reviewer’s comment #22:

Line 203: delete ‘it has to be concluded’

Response to the reviewer’s comment #22:

As proposed, we deleted this phrase.

Reviewer’s comment #23:

206: Weak associations…not sure these are able to be called associations

Response to the reviewer’s comment #23:

The reviewer is right and we rephrased it to “tendencies”.

Reviewer’s comment #24:

208: This sentence is confusing as currently configured. Perhaps it should be, “However, significance is low and not consistent for both species?”

Response to the reviewer’s comment #24:

We thank the reviewer for this helpful suggestion.

Reviewer’s comment #25:

217: first use of effect (as a verb) should be changed to affect, second use of effect (as a noun) is correct

Response to the reviewer’s comment #25:

Thank you for spotting this typo. We corrected it.

Reviewer’s comment #26:

217: change, “it seems justified to conclude that” – is just too wordy

Response to the reviewer’s comment #26:

The sentence was shortened.

Reviewer’s comment #27:

224: sentence needs to shortened

Response to the reviewer’s comment #27:

We separated this aspect into two sentences.

Reviewer’s comment #28:

227 on: not clear what is meant by these two sentences

Response to the reviewer’s comment #28:

As requested, we have clarified in more detail, that the frequency of previous therapeutic interventions might be a difficult to control source of bias, because PCR becomes negative if active infection is therapeutically cured, leading to a lower prevalence estimation of previous infections. To estimate the relevance of this bias, parallel serological assessment of the samples as performed in a previous study would have been an option, because serology also indicates previous, already successfully cured infections.

Reviewer’s comment #29:

I think the discussion also needs to put the findings in better context. What are the prevalences of these two diseases in Ghana and/or the regions of Ghana where the samples were taken? Were the results a surprise or in line with what was expected?

Response to the reviewer’s comment #29:

We agree that a comment on the matching of observations and expectations should be made. So, we have now stated just at that beginning of the discussion (new second sentence) that as expected for Ghana, the majority of individuals was free of active schistosomiasis as indicated by serum PCR and no hints for DNA of systemically circulating leishmaniae were found. Reference 47, in which a similarly low prevalence of schistosomiasis was shown, had already been quoted in the subsequent sentence.

Reviewer’s comment #30:

234: Western African Ghana: this is used in the abstract and here. Capitalization of all three words make it seem that this is to differentiate it from some other country with Ghana in the name. Isn’t there only one Ghana?Using that name was sufficient throughout the rest of the manuscript? Perhaps say, “Ghana, in west Africa” in the Abstract but no need to do it again. Repetitive of discussion

Response to the reviewer’s comment #30:

We changed the wording accordingly.

Reviewer’s comment #31:

What kind of power did this study have? Can the authors do a power analysis to determine what kind of study would be needed to detect a weak association?

Response to the reviewer’s comment #31:

As summarized in the new 6th limitation (please also see our response to your sixteenth comment), the study was exploratory and hypothesis-forming. So, as already stated in the third limitation, no sample size calculation had been made, because the sample size was defined by the numbers of available residual sample materials. In the description of the sixth limitation, we have clarified that confirmatory studies should be larger sized and sufficiently powered, by conducting sample size calculation as suggested by you. The required sample size, however, will depend on the chosen endpoints of such confirmatory studies. Accordingly, no general recommendation for sample sizes fitting all potentially chosen endpoints can be provided, but the data from our study can be used for specifically-adapted sample size calculations as now stated in the last sentence of the sixth limitation at the end of the discussion.

We thank the reviewers for the thorough review which helped improving our manuscript and hope to have addressed all issues raised appropriately.

Sincerely yours,

Kirsten Alexandra Eberhardt, on behalf of the authors

Reviewer 2 Report

The authors have used a little over 1000 samples taken previously for another study in which HIV status, viral load, CD4+ counts and CD4/CD8 ratios were determined. They used qPCR to determine the presence of Leishmania (presumably cutaneous because visceral is not known in Ghana), and Schistosoma mansoni and Schistosoma haemotobium.

The study seemed predicated on the hypothesis that there may be a relationship between HIV status and infection with these parasites, but it’s not clear what the rationale is for that. There is an emphasis that it has not been examined before. However, if schistosomes commonly infect plenty of people who are not immune-compromised, why would a strong association be expected?

In addition, the HIV- group was small. Can such a data set really test for an association?

Introduction

Two studies show high relapse rates of visceral leishmaniasis in HIV+ people. No studies of cutaneous. Isn’t relapse rate a different question than association with immunocompromised status? Did these studies find no association with visceral leishmaniasis absence/presence and immune status?

Infection rates in Ghana go up during rainy season. Were these samples gathered in the rainy season? Seems like the relevance of this statement needs to be dealt with once it is included.

68-69: Change to, “However, education and increased awareness cannot…”

70: Change to, “Of note,…”

72: Change to, “Similar to leishmaniasis, little is known…”

One study found an overall increase of intestinal parasite carriage. What does this mean? (other parasites, intensity of schistosomes?, intensity of other parasites?). Another study detected increased co-incidence rates of urogenital schisto and STDs.

What about other countries? Maybe the rationale for the study would be more clear if there were other studies in other populations that found significant associations? While I appreciate that human populations are very different in genetics and culture, especially in Africa, I’m not sure why only literature with data from Ghana are included.

96: Isn’t this the basis of the qPCR test itself? Not sure of the purpose of this statement

99: tested for correlation? You don’t correlate data, you test for a correlation.

M&M

105: why aren’t the numbers of each group given here in the text? The HIV+ group was small

110: how was the HIV viral load measured in the previous studies, by qPCR?

111: how long had the samples been stored prior to analysis? Could this be a factor in the mostly negative results? Were the extractions new for this study?

2.4: shouldn’t these ethics statements just be included at the end of the paper?

Results

Overall, I am nervous about the number of statistical comparisons being made and that most have at least one category with a small sample size. If one makes enough statistical comparisons, sooner or later there will be one or some that appear significant. It’s not clear that has been considered or corrected for.

I also wonder if it would be more appropriate to use some kind of statistical model to evaluate this data. I am not a statistician, but it seems like this is now often recommended.

Was the viral load data inclusive of HIV negative patients, or were only HIV+ patient data used?

Tables 1 and 2: It’s not clear what the numbers are underneath the p-values.

Discussion

Line 199: ‘Similar as suggested by’ is awkward. Try ‘Like a previous study’, or ‘As similarly seen in a previous study’.

Line 202: which got lost is not appropriate terminology

Line 203: delete ‘it has to be concluded’

206: Weak associations…not sure these are able to be called associations

208: This sentence is confusing as currently configured. Perhaps it should be, “However, significance is low and not consistent for both species?”

217: first use of effect (as a verb) should be changed to affect, second use of effect (as a noun) is correct

217: change, “it seems justified to conclude that” – is just too wordy

224: sentence needs to shortened

227 on: not clear what is meant by these two sentences

I think the discussion also needs to put the findings in better context. What are the prevalences of these two diseases in Ghana and/or the regions of Ghana where the samples were taken? Were the results a surprise or in line with what was expected? 

Conclusions

234: Western African Ghana: this is used in the abstract and here. Capitalization of all three words make it seem that this is to differentiate it from some other country with Ghana in the name. Isn’t there only one Ghana?Using that name was sufficient throughout the rest of the manuscript? Perhaps say, “Ghana, in west Africa” in the Abstract but no need to do it again.

Repetitive of discussion

What kind of power did this study have? Can the authors do a power analysis to determine what kind of study would be needed to detect a weak association?

Overall, I feel that this paper has too many issues to be accepted as it is. I do, however, encourage the authors to work toward getting it published, taking into account this review and any others. While a lot of the results were 'negative', I do believe negative findings should be reported as much as possible. However, one does have to make extra effort to provide the context and the reasons that the results are useful or interesting.

Author Response

(The authors gave the same response as above.)

Round 2

Author Response

Dear Reviewer, Dear Editor,

Thank you very much for your time reviewing our manuscript “Screening for Schistosoma spp. and Leishmania spp. DNA in serum of Ghanaian patients with acquired immunodeficiency” a second time. We highly appreciate your constructive suggestions and are pleased to send you a revised version of our article. Please find our responses below.

Reviewer 2

Reviewer’s comment #1:

You have made the hypothesis more specific and directed to circulating parasite DNA. That being the case, then I would expect that you would include some mention of other studies that have proposed or found this correlation. At the same time, you did not address my concern that since schistosomes quite easily and commonly infect people who have uncompromised immune systems, why would you expect any difference between the two groups?

Response to the reviewer’s comment #1:

We have now addressed these points in the new last two sentences of the introduction and provided respective references.

Reviewer’s comment #2:

It is now better, but you still did not make it clear whether these studies saw a rise in prevalence or intensity of other enteric parasite infections.

Response to the reviewer’s comment #2:

As requested, it has been clarified that “the proportion of intestinal carriage […]” was meant (Introduction, third paragraph, second sentence).

Reviewer’s comment #3:

There are many studies that have considered HIV status with schistosome and/or leishmania infection elsewhere in Africa. It is unacceptable to say that there have not been studies. Perhaps that is true for Ghana, but you must at least say something about the topic in other areas of Africa. A full summary of the literature is not expected, but at least a few references should be cited. One start is this review: https://www.ncbi.nlm.nih.gov/pmc/articles/PMC7124894/. This literature is important for the introduction and as context in your discussion of your results.

Response to the reviewer’s comment #3:

Next to the changes in response to your point #1, we have now discussed our findings in relation to previous results at the end of the discussion paragraphs 1-3 as well as at the end of the conclusion paragraph. Respective references have been provided.

Reviewer’s comment #4:

I think the issue was created (and solved) by the spacing in the table.

Response to the reviewer’s comment #4:

Thank you for the confirmation that the spacing issue had been solved.

Reviewer’s comment #5:

What you have added is helpful, but still very general and vague. Connected to #1 and #9, you need to provide some context for the reader in the introduction and the discussion.

Response to the reviewer’s comment #5:

As detailed in response to your comments #1 and #4, this context has now been provided.

We thank the reviewer for the thorough review which helped improving our manuscript and hope to have addressed all issues raised appropriately.

Sincerely yours,

Kirsten Alexandra Eberhardt, on behalf of the authors
